# Feasibility of Seated Stepping and Handshaking as a Cardiopulmonary Exercise Testing: A Pilot Study

**DOI:** 10.3390/jcm12062140

**Published:** 2023-03-09

**Authors:** Kyosuke Imashiro, Yasuko Nishioka, Kenzo Teramura, Hiromi Hashimoto, Hiroki Kimura, Naoya Tanabe, Yasuhiro Taniguchi, Koya Nakai, Yasunori Umemoto, Tomoyuki Ito, Fumihiro Tajima, Yasuo Mikami

**Affiliations:** 1Department of Rehabilitation Medicine, Wakayama Medical University, 811-1 Kimiidera, Wakayama 641-8509, Japan; 2Department of Rehabilitation, Kyoto Konoe Rehabilitation Hospital, 26 Yoshidakonoecho, Sakyo-ku, Kyoto 606-8315, Japan; 3Department of Rehabilitation Medicine, Graduate School of Medical Science, Kyoto Prefectural University of Medicine, 465 Kajii-cho, Kawaramachi-Hirokoji, Kamigyo-ku, Kyoto 602-8566, Japan

**Keywords:** aerobic exercise, cardiopulmonary exercise test, cycle ergometer, submaximal exercise test

## Abstract

Cardiopulmonary function is usually assessed by cardiopulmonary exercise testing (CPX) using a cycle ergometer (CE-CPX) or a treadmill, which is difficult in patients with lower extremity motor dysfunction. A stepping and handshaking (SHS) exercise has been developed that can be performed safely and easily while sitting on a chair. This study compared peak oxygen uptake (peak $\dot{V}O_{2}$) between CE-CPX and SHS-CPX in healthy adults and investigated the safety and validity of SHS-CPX. Twenty young adults (mean age 27.8 ± 4.4 years) were randomly assigned to perform CE-CPX or SHS-CPX, with the other test to follow 1–2 weeks later. The peak $\dot{V}O_{2}$, respiratory exchange ratio (RER), peak heart rate, blood pressure, and test completion time were compared between CE-CPX and SHS-CPX. All subjects completed the examination and met the criteria for peak $\dot{V}O_{2}$. SHS-CPX and CE-CPX showed a strong correlation with peak $\dot{V}O_{2}$ (r = 0.85, *p* < 0.0001). The peak $\dot{V}O_{2}$ (40.4 ± 11.3 mL/min/kg vs. 28.9 ± 8.0 mL/min/kg), peak heart rate (190.6 ± 8.9 bpm vs. 172.1 ± 12.6 bpm), and test completion time (1052.8 ± 143.7 s vs. 609.1 ± 96.2 s) were significantly lower in the SHS-CPX (*p* < 0.0001). There were no adverse events. The peak $\dot{V}O_{2}$ with SHS-CPX was equivalent to about 70% of that with CE-CPX despite the exercise being performed in a sitting position, suggesting its suitability as a submaximal exercise test.

## 1. Introduction

Healthy life expectancy is becoming important with increasing longevity [1]. Decreased cardiopulmonary function is known to have an adverse impact on quality of life [2] and to be associated with increased early mortality [3]. The gold standard for the assessment of cardiopulmonary function is the cardiopulmonary exercise test (CPX) using a treadmill or cycle ergometer (CE) [4,5,6]. However, this approach may have limited application if individuals are not able to meet the criteria for maximal exertion due to fatigue, motivation [6], or physical limitations [7]. Therefore, submaximal exercise load testing is being investigated as an alternative to CPX [8]. A submaximal exercise test estimates aerobic capacity based on the assumption that there is a linear relationship between oxygen uptake and heart rate (HR) within a fixed range [9]. Furthermore, compared with CPX, a submaximal exercise test is a safer way of assessing cardiopulmonary function. Several submaximal exercise tests have been developed, including the incremental sit-to-stand exercise [10], total body recumbent stepper submaximal exercise test [11], stepping ergometer [12], step exercise [13], and Astrand cycle tests [14]. However, all of these tests involve the use of pedals and standing up and are unsuitable for people with motor dysfunction in the lower extremities. A stepping and handshaking (SHS) exercise has been developed that can be performed safely and easily while sitting in a chair. To the best of our knowledge, there are no reports on CPX using the SHS. We have devised a method for the assessment of cardiopulmonary function using the SHS test. This pilot study was designed to compare peak $\dot{V}O_{2}$ during SHS-CPX with that during CE-CPX and to evaluate the safety and usefulness of SHS-CPX in healthy young adults who were able to undergo both tests.

## 2. Materials and Methods

The study protocol was approved by the Ethics Review Committee of Wakayama Medical University (approval number 2737) and performed in accordance with the principles laid down in the Declaration of Helsinki. The study participants were 20 healthy adult volunteers (14 male, 6 female; mean age 27.8 ± 4.4 years) who were recruited between April 2020 and October 2020. All participants were fully informed of the purpose of the study and its procedures and provided both written and verbal informed consent before the tests were performed.

The study inclusion criteria were non-smoking status, age 20 years or older, and no history of musculoskeletal, cardiopulmonary, metabolic, or psychiatric conditions that would interfere with the ability to exercise.

The physical characteristics of the study participants are shown in Table 1. Participants were randomized to perform CE-CPX or SHS-CPX and the alternative test 1 week later. All tests were performed in a rehabilitation room at an ambient temperature of 28 °C between 5 p.m. and 7 p.m. All participants completed the study.

### 2.1. CE-CPX

The CE-CPX protocol employed in this study was partially modified from a previous study conducted on healthy young adults by Ashley et al. [15]. Peak $\dot{V}O_{2}$ was measured by the incremental loading method using a cycle ergometer (915E, Monark, Varberg, Sweden). After resting for 3 min, the subject increased the load in the order of 0, 50, and 100 W at 3 min intervals, and then gradually increased the load by 20 W at 1 min intervals. The test was terminated when the pedal rotation speed could not be maintained at 60 rpm.

### 2.2. SHS-CPX

The SHS was performed with the subject sitting on a chair (seat height 42 cm) without a backrest. The subject performed a stepping exercise that consisted of swinging the arms and raising the thighs while rotating the trunk in time to the tempo of a metronome (Figure 1). The exercise load was started at 80 beats/min, increased by 10 beats/min at 1 min intervals up to 120 beats/min, and then gradually increased by 5 beats/min. The examination ended when the limb movements were no longer in sync with the metronome. From a physiological standpoint, if the load increases at intervals of one minute or less, it is considered a ramp load [16]. In addition, SHS-CPX was designed to be a protocol that could be completed in 8–12 min [17] after a number of pre-experiments.

### 2.3. Outcome Measures

The peak $\dot{V}O_{2}$ and respiratory exchange ratio (RER) were calculated using a portable breath gas analyzer (MetaMax 3B, CORTEX, Leipzig, Germany). The average value was calculated breath-by-breath at 15 s intervals, and the average value for the 15 s immediately before the end of the test was taken as the peak $\dot{V}O_{2}$. HR was recorded at 1 min intervals on an electrocardiographic monitor (BMS-2401, Nihon Kohden, Tokyo, Japan) worn by the subject. Blood pressure was measured using a sphygmomanometer (Eremano 2 ES-H56; Terumo, Tokyo, Japan) immediately before and after the examination. The time taken to complete the test was measured with a stopwatch.

### 2.4. Statistical Analysis

The study data are shown as the mean ± standard deviation. The normality of the data was examined using the Shapiro–Wilk test. The Pearson’s product-moment correlation coefficient was used to determine the correlation of the peak $\dot{V}O_{2}$ values with each type of CPX. In addition, Bland–Altman analysis was used to assess the consistency of the data on peak $\dot{V}O_{2}$ for each type of CPX. The average values for peak $\dot{V}O_{2}$, RER, HR, blood pressure, and completion time for each type of CPX were compared using paired *t*-tests. All statistical analyses were performed using GraphPad Prism 7 (GraphPad Software Inc., La Jolla, CA, USA), with a significance level of less than 5%.

## 3. Results

All participants were able to complete the test, and there were no adverse events. The peak $\dot{V}O_{2}$ during CE-CPX was positively correlated with that during SHS-CPX (r = 0.85, *p* < 0.0001, Figure 2). However, the Bland–Altman analysis resulted in a fixed error (Figure 3). The values for peak $\dot{V}O_{2}$, peak HR, RER, and test completion time were significantly lower for the SHS-CPX than for the CE-CPX. The peak $\dot{V}O_{2}$ measured by SHS-CPX was 87.2 ± 9.8% of that measured by CE-CPX. There was no significant difference in resting HR or blood pressure (Table 2).

## 4. Discussion

This pilot study measured peak $\dot{V}O_{2}$ using the SHS-CPX. All study participants were healthy young adults and able to complete the test. There were no adverse events. There was a significant strong correlation (r = 0.85) of the peak $\dot{V}O_{2}$ between SHS-CPX and CE-CPX. This value is higher than that found in a previous study that examined the relationship between the incremental shuttle walk test and peak $\dot{V}O_{2}$ on CE-CPX (r ≥ 0.70) [18]; it may be explained by the fact that SHS-CPX is an exercise that required the muscles of the upper limbs, lower limbs, and trunk to move in a rhythmic fashion, whereas CE-CPX only requires the rhythmic movement of the muscles of the lower limbs, and, that in both cases the exercise load and rhythm are externally controlled. This indicates that SHS-CPX is a useful method for assessing aerobic capacity.

The peak $\dot{V}O_{2}$ was significantly lower for SHS-CPX than for CE-CPX. However, peak $\dot{V}O_{2}$ was significantly lower in SHS-CPX than in CE-CPX, with peak $\dot{V}O_{2}$ measurements (28.9 ± 8.0 mL/min/kg) being almost 70% of CE-CPX (40.4 ± 11.3 mL/min/kg). In a previous study, the predicted peak $\dot{V}O_{2}$ obtained by the Chester step test was close to 90% of the measured peak $\dot{V}O_{2}$ [19], indicating a high correlation [20]. However, the present study is the first to report that the peak $\dot{V}O_{2}$ measured using an unweighted exercise method while seated on a chair is approximately 70% of the CE-CPX, again indicating a high correlation. Furthermore, the time required to perform SHS-CPX is about half that needed to perform CE-CPX. On the other hand, as shown in the Bland–Altman analysis, the SHS-CPX has a fixed error and the SHS-CPX may be difficult to drive at the same load as the CE-CPX. The validity of the SHS-CPX as a submaximal exercise test needs to be verified in the future.

This study has some limitations, including a relatively small sample size, a single-center design, and study subjects who were healthy young adults. A larger sample size is needed to generalize the results of this pilot study. Furthermore, reproducibility could not be examined. Further studies in patients with lower limb motor dysfunction, such as osteoarthritis or rheumatoid arthritis, are needed to determine whether SHS-CPX can be applied in a clinical setting.

## 5. Conclusions

The peak $\dot{V}O_{2}$ for the SHS-CPX was strongly correlated with that for the CE-CPX and could measure about 70% of the exercise load when compared with that for the CE-CPX in a short time. This suggested the feasibility of SHS-CPX as a submaximal exercise test. The SHS-CPX does not require a large space and can be performed while seated in a chair. Our findings suggest that the SHS-CPX is a safe and useful method for the assessment of aerobic capacity. We will need to carefully determine whether SHS-CPX is applicable to patients with motor dysfunction and the elderly as we continue to add cases.

## Figures and Tables

**Figure 1 jcm-12-02140-f001:**
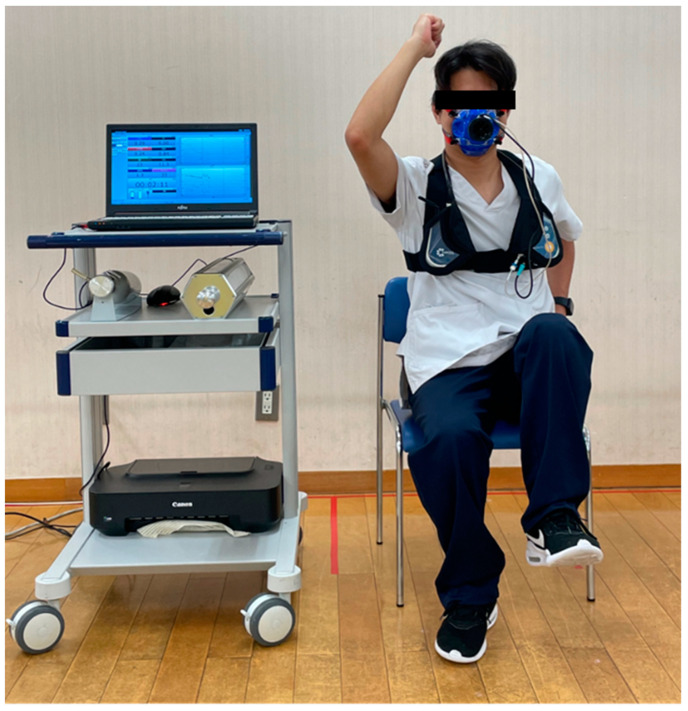
Photograph showing stepping and handshaking exercise in the sitting position (SHS) as a cardiopulmonary test.

**Figure 2 jcm-12-02140-f002:**
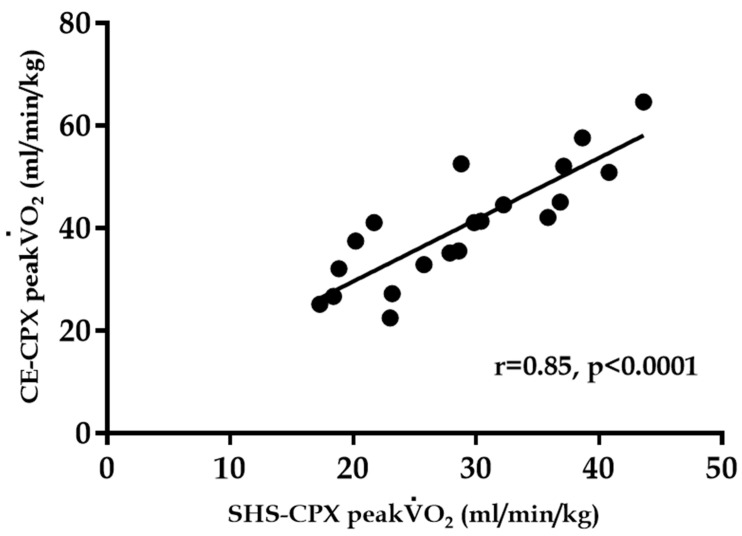
Graph showing the correlation between peak $\dot{V}O_{2}$ obtained during CE-CPX and that obtained during SHS-CPX. CE-CPX: cycle ergometer cardiopulmonary testing; SHS-CPX: sitting and handshaking exercise cardiopulmonary testing; $\dot{V}O_{2}$: oxygen uptake.

**Figure 3 jcm-12-02140-f003:**
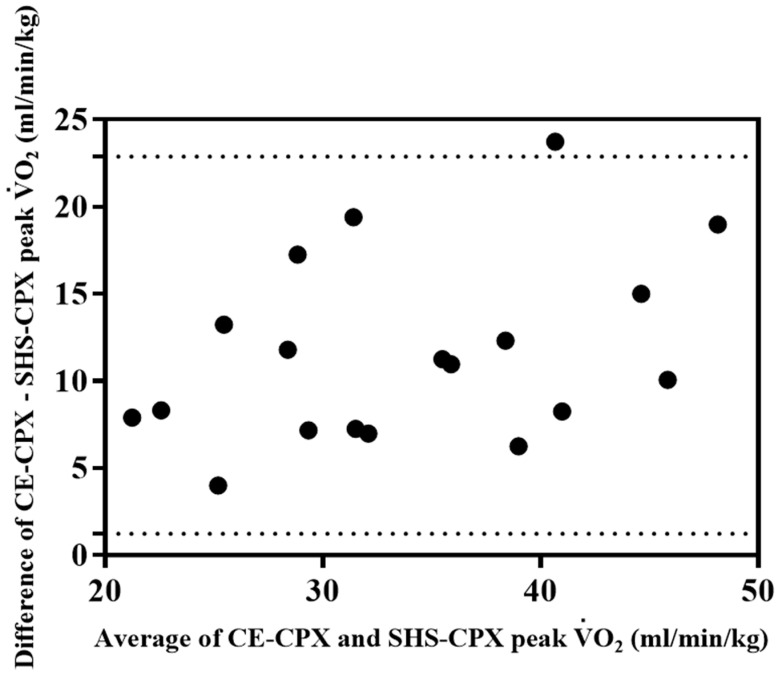
Bland–Altman analysis of the correlation between the CE-CPX peak $\dot{V}O_{2}$ and SHS-CPX peak $\dot{V}O_{2}$. CE-CPX: cycle ergometer cardiopulmonary testing; SHS-CPX: sitting and handshaking exercise cardiopulmonary testing; $\dot{V}O_{2}$: oxygen uptake.

**Table 1 jcm-12-02140-t001:** Anthropometric characteristics of the study population according to sex.

	Total (n = 20)	Men (n = 14)	Women (n = 6)
Age (years)	27.8 ± 4.4	27.5 ± 4.8	28.5 ± 3.8
Height (cm)	169.1 ± 6.3	171.9 ± 5.1	162.5 ± 3.3
Body weight (kg)	63.5 ± 11.6	66.7 ± 12.2	56.0 ± 4.7
Body mass index (kg/m^2^)	22.1 ± 3.2	22.5 ± 3.7	21.2 ± 1.6

Data are presented as the mean ± standard deviation.

**Table 2 jcm-12-02140-t002:** Measurements obtained at peak exercise and at rest.

	CE-CPX	SHS-CPX	*p*-Value	95% CI
Resting cardiopulmonary function				
Heart rate (bpm)	79.3 ± 9.4	75.1 ± 9.7	0.0918	−9.15 to 0.75
Systolic blood pressure (mmHg)	125.6 ± 13.2	122.3 ± 13.5	0.1378	−7.876 to 1.176
Diastolic blood pressure (mmHg)	81.4 ± 11.8	81.3 ± 9.7	0.9530	−5.411 to 5.111
Peak cardiopulmonary function				
Peak $\dot{V}O_{2}$ (mL/min/kg)	40.4 ± 11.3	28.9 ± 8.0	<0.0001	−14.48 to −7.29
Respiratory exchange ratio	1.17 ± 0.04	1.09 ± 0.05	<0.0001	−0.105 to −0.050
Heart rate (bpm)	190.6 ± 8.9	172.1 ± 12.6	<0.0001	−24.96 to −11.84
Systolic blood pressure (mmHg)	143.6 ± 31.8	143.6 ± 16.6	0.9937	−13.19 to 13.09
Diastolic blood pressure (mmHg)	79.5 ± 18.2	81.8 ± 11.7	0.4777	−4.347 to 8.947
Completion time (s)	1052.8 ± 143.7	609.1 ± 96.2	<0.0001	−506.9 to −380.6
Peak workload (watt)	202.5 ± 60.1	-	-	-
Peak workload (beat/min)	-	133.3 ± 7.9	-	-

Data are presented as the mean ± standard deviation. CE-CPX: cycle ergometer cardiopulmonary exercise; CI: confidence interval; SHS-CPX: stepping and handshaking cardiopulmonary exercise; $\dot{V}O_{2}$: oxygen uptake.

## Data Availability

The datasets generated and/or analyzed during the current study are not publicly available due to institutional privacy policy but are available from the corresponding author upon reasonable request.

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
