# Peer review of "Feasibility of Seated Stepping and Handshaking as a Cardiopulmonary Exercise Testing: A Pilot Study"

_jcm, 2023, doi:10.3390/jcm12062140_

Round 1
Reviewer 1 Report
This is an interesting proposal of a method to allow for cardiopulmonary test in patients with lower extremity motor dysfunction. However given the choice of a very homogeneous small population of young healthy subjects the conclusion of strong correlation are not sufficiently convincing.
I would suggest 1) to increase the variability by including a wider range of subjects/patients of different age, gender, fitness, etc; 2) increase the sample size; present a Bland & Altman plot that will allow to understand easily whether the systematic yunder/overestimation is constant throughout a wide range of values
Author Response
Dr. Emmanuel Andrès Editor-in-Chief
Journal of Clinical Medicine
1 March 2023
Dear Dr Andrès
On behalf of my co-authors, I thank you for the opportunity to revise our manuscript (ID : jcm-2225008), titled “Feasibility of seated stepping and handshaking as a cardiopulmonary exercise testing: a pilot study”. The reviewers’ comments were very insightful and helped us in revising and improving our manuscript.
We have carefully addressed all the reviewers’ comments in our revised manuscript. We hope that our responses and revisions have adequately addressed the reviewers’ concerns and that the revised manuscript will now meet the high standards required for publication in your esteemed journal.
The manuscript has been rechecked and the necessary changes have been made in accordance with the reviewers’ suggestions. The responses to all comments have been prepared and given below. We have applied the “Track Changes” feature of MS Word to highlight the revisions in our updated manuscript.
Thank you for your consideration. We look forward to hearing from you.
Sincerely,
Yasuko Nishioka, MD, PhD
Department of Rehabilitation Medicine, Wakayama Medical University, 811-1 Kimiidera, Wakayama, 640-8509, Japan
Telephone: +81-73-447-0664
Fax: +81-73-446-6475
E-mail: kopandap627@yahoo.co.jp
Response to Reviewer 1 Comments
Manuscript ID: jcm-2225008
Type of manuscript: Brief Report
Title: Feasibility of seated stepping and handshaking as a cardiopulmonary exercise testing: a pilot study
Point-by-point responses to the reviewers’ comments
Dear Editor,
We would like to thank the Reviewers and Editors for their time and effort in reviewing our manuscript and for providing comments and suggestions, which have considerably helped us improve our manuscript. We have answered each of your points below and hope that our responses and revisions address all your comments.
Point 1: I would suggest to increase the variability by including a wider range of subjects/patients of different age, gender, fitness, etc.
Response 1: This study was conducted only with healthy young adults who could safely perform both CE-CPX and SHS-CPX. Gender was a mixture of men and women.
Point 2: I would suggest to increase the sample size; present a Bland & Altman plot that will allow to understand easily whether the systematic yunder/overestimation is constant throughout a wide range of values.
Response 2: The sample size was increased from 11 to 20 participants.
When doing the Bland & Altman plot, there was a fixed error in peak VO2 for CE-CPX and SHS-CPX. This is because the peak VO2 of SHS-CPX is about 70% of that of CE-CPX.

Reviewer 2 Report
I reviewed with interest the manuscript by Imashiro et al. "Feasibility of seated stepping and handshaking as a cardio-pulmonary exercise testing: a pilot study". In this article, the authors reasonably note that a proportion of patients are unable to perform a conventional cardiopulmonary test due to lower limb motor dysfunction such as osteoarthritis or rheumatoid arthritis. Therefore, in such cases, alternative exercise tests (other than the traditional treadmill or bicycle ergometer) should be considered. The authors proposed a new version of such a test - stepping and handshaking (SHS) exercise. This article compares this test with the cardiopulmonary test in terms of its effect on maximum oxygen consumption. Indeed, a high degree of VO2max correlation was shown when performing these tests.
However, when reviewing the manuscript, I had questions and comments to which I would like to receive answers from the authors.
1. The authors used a rather original CE-CPX protocol. I would like to know what previous studies have assessed peak VO2 with this test.
2.I also have questions about the SHS-CPX test. What is the test protocol used based on, in particular, how was the duration of the load steps determined? Section 2.2. SHS-CPX states "The exercise load was started at 80 beats/min, increased by 10 beats/min at 1-min intervals up to 120 beats/min, and then gradually increased by 5 beats/min until exercise was defined by cessation of operation of the metronome". A logical question - when did the metronome stop? What was the criterion for stopping him? Secondly, what level of load in beats/min did the subjects achieve?
3. What formula did the authors use to calculate the predicted maximum HR?
4. According to table 2 - two parameters have the same name - Peak VO2. This may cause misunderstanding among readers, it is necessary to eliminate the ambiguity.
5. In the Discussion section, the authors state that "Quinn et al. [20] reported that exercise equivalent to 70% of VO2max was associated with significantly higher oxygen excess consumption post-exercise, suggesting that the exercise load is significantly demanding". In my opinion, it is incorrect to cite this work as an example, since, firstly, the duration of exercise in that study was 20, 40 or 60 minutes, and the duration of the SHS-CPX test was no more than 10 minutes. In addition, in that study, the entire load was at the level of 70% of VO2max, and with the SHS-CPX test, the duration of the load close to this level can be 2-3 minutes.
6. The possibility of clinical use of the SHS-CPX test is questionable. Apparently, when performing this test, the requirements for joint mobility are also high, as well as additionally for the speed of performing this test. All this can become an obstacle to the adequate performance of this test by persons with impaired motor function.
7. Noteworthy is the use by the authors of references to old studies: 12 out of 20 references are more than 10 years old, and 5 of them belong to the last century. Unwittingly doubts creep in the relevance of this study. At the same time, the authors did not consider more recent publications on this issue. I would recommend that they consider, for example, the recent review by Cuenca‑Garcia et al (1).
References:
1. Cuenca-Garcia M, Marin-Jimenez N, Perez-Bey A, Sánchez-Oliva D, Camiletti-Moiron D, Alvarez-Gallardo IC, Ortega FB, Castro-Piñero J. Reliability of Field-Based Fitness Tests in Adults: A Systematic Review. Sports Med. 2022 Aug;52(8):1961-1979. doi: 10.1007/s40279-021-01635-2.
Author Response
Dr. Emmanuel Andrès Editor-in-Chief
Journal of Clinical Medicine
1 March 2023
Dear Dr Andrès
On behalf of my co-authors, I thank you for the opportunity to revise our manuscript (ID : jcm-2225008), titled “Feasibility of seated stepping and handshaking as a cardiopulmonary exercise testing: a pilot study”. The reviewers’ comments were very insightful and helped us in revising and improving our manuscript.
We have carefully addressed all the reviewers’ comments in our revised manuscript. We hope that our responses and revisions have adequately addressed the reviewers’ concerns and that the revised manuscript will now meet the high standards required for publication in your esteemed journal.
The manuscript has been rechecked and the necessary changes have been made in accordance with the reviewers’ suggestions. The responses to all comments have been prepared and given below. We have applied the “Track Changes” feature of MS Word to highlight the revisions in our updated manuscript.
Thank you for your consideration. We look forward to hearing from you.
Sincerely,
Yasuko Nishioka, MD, PhD
Department of Rehabilitation Medicine, Wakayama Medical University, 811-1 Kimiidera, Wakayama, 640-8509, Japan
Telephone: +81-73-447-0664
Fax: +81-73-446-6475
E-mail: kopandap627@yahoo.co.jp
Response to Reviewer 2 Comments
Manuscript ID: jcm-2225008
Type of manuscript: Brief Report
Title: Feasibility of seated stepping and handshaking as a cardiopulmonary exercise testing: a pilot study
Point-by-point responses to the reviewers’ comments
Dear Editor,
We would like to thank the Reviewers and Editors for their time and effort in reviewing our manuscript and for providing comments and suggestions, which have considerably helped us improve our manuscript. We have answered each of your points below and hope that our responses and revisions address all your comments.
Point 1: The authors used a rather original CE-CPX protocol. I would like to know what previous studies have assessed peak VO2 with this test.
Response 1: The CE-CPX protocol employed in this study is a modification of a previous study conducted on healthy young adults.
The text in lines 135-136 has been revised and references [15] have been added.
Point 2: I also have questions about the SHS-CPX test. What is the test protocol used based on, in particular, how was the duration of the load steps determined? Section 2.2. SHS-CPX states "The exercise load was started at 80 beats/min, increased by 10 beats/min at 1-min intervals up to 120 beats/min, and then gradually increased by 5 beats/min until exercise was defined by cessation of operation of the metronome". A logical question - when did the metronome stop? What was the criterion for stopping him? Secondly, what level of load in beats/min did the subjects achieve?
Response 2: The following is a response regarding the SHS-CPX testing protocol.
Agostoni [16] et al. reported that, from a physiological standpoint, if the load increased at intervals of 1 min or less, it was considered a ramp load, and the loading step duration was increased progressively every 1 min.
Levett [17] et al. reported that an 8-12 minute exercise load test is ideal, and SHS-CPX was designed to be a protocol that could be completed in 8-12 minutes with a number of pre-experiments.
I added text to lines 298-301 of the text and added [16][17] to the references.
I did not include enough in the protocol regarding the end of the test. The test was terminated when the limb movements were no longer in sync with the metronome tempo.
I changed the sentence in lines 297-298 of the text.
The average subject load in the SHS-CPX was 133.8±8.1 (beats/min).
This has been added to Table 2.
Point 3: What formula did the authors use to calculate the predicted maximum HR?
Response 3: The predicted maximal heart rate was "220 - age" as used in the Karvonen Formula.
Point 4: According to table 2 - two parameters have the same name - Peak VO2. This may cause misunderstanding among readers, it is necessary to eliminate the ambiguity.
Response 4: I have changed the notation to ml/min/kg only.
Point 5: In the Discussion section, the authors state that "Quinn et al. [20] reported that exercise equivalent to 70% of VO2max was associated with significantly higher oxygen excess consumption post-exercise, suggesting that the exercise load is significantly demanding". In my opinion, it is incorrect to cite this work as an example, since, firstly, the duration of exercise in that study was 20, 40 or 60 minutes, and the duration of the SHS-CPX test was no more than 10 minutes. In addition, in that study, the entire load was at the level of 70% of VO2max, and with the SHS-CPX test, the duration of the load close to this level can be 2-3 minutes.
Response 5: As you have pointed out, I have removed it from the discussion.
Point 6: The possibility of clinical use of the SHS-CPX test is questionable. Apparently, when performing this test, the requirements for joint mobility are also high, as well as additionally for the speed of performing this test. All this can become an obstacle to the adequate performance of this test by persons with impaired motor function.
Response 6: As you mentioned, it is difficult to perform the test when there is joint range of motion limitation in the hip or shoulder joints. We are mainly assuming patients who are unable to ride an aerobike or who are unstable in the standing position. We believe that future work is needed to verify whether the test can be applied to patients with motor dysfunction and the elderly.
Point 7: Noteworthy is the use by the authors of references to old studies: 12 out of 20 references are more than 10 years old, and 5 of them belong to the last century. Unwittingly doubts creep in the relevance of this study. At the same time, the authors did not consider more recent publications on this issue. I would recommend that they consider, for example, the recent review by Cuenca‑Garcia et al (1).
Response 7: As you indicated, we have removed all references from the last century and replaced them with more recent references.

Reviewer 3 Report
I read with interest the study by Kyosuke Imashiro et al.
This is a very well written manuscript where the feasibility of seated stepping and handshaking as a cardiopulmonary exercise testing is assessed in this pilot study.
As this is a pilot study all my remarks are in the context of the design of the study.
It is widely accepted that a fairly wide proportion of patients undergoing CPX may not be physically able to have this due to underlying musculoskeletal conditions which may be prohibitive.
The whole rationale of this study is based on the premise that an alternative method of exercising may lead to similar and reproducible results.
The authors suggest the seated stepping and handshaking method which they describe in almost exhaustive detail in the text and should be commended on this.
I have a few remarks on this paper:
- The participants were young and apparently healthy with low BMI and as expected both tests were completed by the 11 participants. Nonetheless, and although there is a fairly good correlation of peakVO2 between the two methods, there was statistically significant difference between the two tests. This implies that one of the major determinants in assessing the cardiopulmonary condition of a person may not be sufficiently derived by the method suggested by the authors.
- It is not clear in the text which is the determinant of stopping the metronome and hence the exercise part of the test. I assume it is related to the achievement of maximal predicted heart rate but not unequivocally documented in the text.
- This is a very small study, evidently a pilot study with healthy participants and all observations should be presented extremely cautiously. I believe that the Conclusions section should be toned down a lot.
Overall, I believe that this is an interesting and novel concept with some merit.
However, without evidence of reproducibility of the results particularly in a proper cohort of patients, the observations should be reproduced in a more appropriate population. I would suggest an equally small number of patients who would normally be referred for CPX assessment and this would still be in the context of a pilot study.
Author Response
Dr. Emmanuel Andrès Editor-in-Chief
Journal of Clinical Medicine
1 March 2023
Dear Dr Andrès
On behalf of my co-authors, I thank you for the opportunity to revise our manuscript (ID : jcm-2225008), titled “Feasibility of seated stepping and handshaking as a cardiopulmonary exercise testing: a pilot study”. The reviewers’ comments were very insightful and helped us in revising and improving our manuscript.
We have carefully addressed all the reviewers’ comments in our revised manuscript. We hope that our responses and revisions have adequately addressed the reviewers’ concerns and that the revised manuscript will now meet the high standards required for publication in your esteemed journal.
The manuscript has been rechecked and the necessary changes have been made in accordance with the reviewers’ suggestions. The responses to all comments have been prepared and given below. We have applied the “Track Changes” feature of MS Word to highlight the revisions in our updated manuscript.
Thank you for your consideration. We look forward to hearing from you.
Sincerely,
Yasuko Nishioka, MD, PhD
Department of Rehabilitation Medicine, Wakayama Medical University, 811-1 Kimiidera, Wakayama, 640-8509, Japan
Telephone: +81-73-447-0664
Fax: +81-73-446-6475
E-mail: kopandap627@yahoo.co.jp
Response to Reviewer 3 Comments
Manuscript ID: jcm-2225008
Type of manuscript: Brief Report
Title: Feasibility of seated stepping and handshaking as a cardiopulmonary exercise testing: a pilot study
Point-by-point responses to the reviewers’ comments
Dear Editor,
We would like to thank the Reviewers and Editors for their time and effort in reviewing our manuscript and for providing comments and suggestions, which have considerably helped us improve our manuscript. We have answered each of your points below and hope that our responses and revisions address all your comments.
Point 1: The participants were young and apparently healthy with low BMI and as expected both tests were completed by the 11 participants. Nonetheless, and although there is a fairly good correlation of peakVO2 between the two methods, there was statistically significant difference between the two tests. This implies that one of the major determinants in assessing the cardiopulmonary condition of a person may not be sufficiently derived by the method suggested by the authors.
Response 1: As you have pointed out, the SHS did not load as well as the CE-CPX. We increased the sample size from 11 to 20 subjects and did a Bland & Altman plot and found a fixed error in peak VO2 for CE-CPX and SHS-CPX. This means that the peak VO2 of SHS-CPX is about 70% of that of CE-CPX, and the loading method of SHS-CPX is not equivalent to that of CE-CPX. We believe that the validity of SHS-CPX as a submaximal exercise test should be examined in the future.
Point 2: It is not clear in the text which is the determinant of stopping the metronome and hence the exercise part of the test. I assume it is related to the achievement of maximal predicted heart rate but not unequivocally documented in the text.
Response 2: I did not include enough in the protocol regarding the end of the test. The test was terminated when the limb movements were no longer in sync with the metronome tempo.
I changed the sentence in lines 297-298 of the text.
Point 3: This is a very small study, evidently a pilot study with healthy participants and all observations should be presented extremely cautiously. I believe that the Conclusions section should be toned down a lot.
Response 3: We have made the correction as you indicated. I have changed lines 628, 631-632 of the conclusion
Point 4: Overall, I believe that this is an interesting and novel concept with some merit.
However, without evidence of reproducibility of the results particularly in a proper cohort of patients, the observations should be reproduced in a more appropriate population. I would suggest an equally small number of patients who would normally be referred for CPX assessment and this would still be in the context of a pilot study.
Response 4: We have made the correction as you indicated. Reproducibility could not be examined in this study. In the future, we will further increase the sample size and examine reproducibility.

Round 2
Reviewer 1 Report
The authors have increased their sample size and recognized the limits of their conclusion in the revised version of the manuscript.
Of note the Bland and Altman plot (for concordance not correlation analysis as stated) shows some relation between the mean and the difference of the measurements (difference seems to increase with the value) which could be of concern and might be verified on large case series with different types of subjects and recognized as a potential bias
Author Response
Dr. Emmanuel Andrès Editor-in-Chief
Journal of Clinical Medicine
6 March 2023
Dear Dr Andrès
On behalf of my co-authors, I thank you for the opportunity to revise our manuscript (ID : jcm-2225008), titled “Feasibility of seated stepping and handshaking as a cardiopulmonary exercise testing: a pilot study”. The reviewers’ comments were very insightful and helped us in revising and improving our manuscript.
We have carefully addressed all the reviewers’ comments in our revised manuscript. We hope that our responses and revisions have adequately addressed the reviewers’ concerns and that the revised manuscript will now meet the high standards required for publication in your esteemed journal.
The manuscript has been rechecked and the necessary changes have been made in accordance with the reviewers’ suggestions. The responses to all comments have been prepared and given below.
Thank you for your consideration. We look forward to hearing from you.
Sincerely,
Yasuko Nishioka, MD, PhD
Department of Rehabilitation Medicine, Wakayama Medical University, 811-1 Kimiidera, Wakayama, 640-8509, Japan
Telephone: +81-73-447-0664
Fax: +81-73-446-6475
E-mail: kopandap627@yahoo.co.jp
Response to Reviewer 1 Comments
Manuscript ID: jcm-2225008
Type of manuscript: Brief Report
Title: Feasibility of seated stepping and handshaking as a cardiopulmonary exercise testing: a pilot study
Point-by-point responses to the reviewers’ comments
Dear Editor,
We would like to thank the Reviewers and Editors for their time and effort in reviewing our manuscript and for providing comments and suggestions, which have considerably helped us improve our manuscript. We have answered each of your points below and hope that our responses and revisions address all your comments.
Point 1: The authors have increased their sample size and recognized the limits of their conclusion in the revised version of the manuscript.
Of note the Bland and Altman plot (for concordance not correlation analysis as stated) shows some relation between the mean and the difference of the measurements (difference seems to increase with the value) which could be of concern and might be verified on large case series with different types of subjects and recognized as a potential bias
Response 1: As you have pointed out, The challenge for the future is to further validate this as a large case series that includes a wide range of subjects and patients with different ages, genders, physical fitness and diseases.

Reviewer 2 Report
The authors answered my questions, made a correction in the text. Questions remained only on the possibility of using this test in real practice, but this does not apply directly to this manuscript.
Author Response
Dr. Emmanuel Andrès Editor-in-Chief
Journal of Clinical Medicine
6 March 2023
Dear Dr Andrès
On behalf of my co-authors, I thank you for the opportunity to revise our manuscript (ID : jcm-2225008), titled “Feasibility of seated stepping and handshaking as a cardiopulmonary exercise testing: a pilot study”. The reviewers’ comments were very insightful and helped us in revising and improving our manuscript.
We have carefully addressed all the reviewers’ comments in our revised manuscript. We hope that our responses and revisions have adequately addressed the reviewers’ concerns and that the revised manuscript will now meet the high standards required for publication in your esteemed journal.
The manuscript has been rechecked and the necessary changes have been made in accordance with the reviewers’ suggestions. The responses to all comments have been prepared and given below.
Thank you for your consideration. We look forward to hearing from you.
Sincerely,
Yasuko Nishioka, MD, PhD
Department of Rehabilitation Medicine, Wakayama Medical University, 811-1 Kimiidera, Wakayama, 640-8509, Japan
Telephone: +81-73-447-0664
Fax: +81-73-446-6475
E-mail: kopandap627@yahoo.co.jp
Response to Reviewer 2 Comments
Manuscript ID: jcm-2225008
Type of manuscript: Brief Report
Title: Feasibility of seated stepping and handshaking as a cardiopulmonary exercise testing: a pilot study
Point-by-point responses to the reviewers’ comments
Dear Editor,
We would like to thank the Reviewers and Editors for their time and effort in reviewing our manuscript and for providing comments and suggestions, which have considerably helped us improve our manuscript. We have answered each of your points below and hope that our responses and revisions address all your comments.
Point 1: The authors answered my questions, made a correction in the text. Questions remained only on the possibility of using this test in real practice, but this does not apply directly to this manuscript.
Response 1: The future challenge is to make the test clinically applicable by including a variety of subjects and patients.

Reviewer 3 Report
I commend the authors for the hard work and reverting with the revised manuscript in short time.
This is a very well written paper with sound analytical methods.
I can still see the novelty in this concept but not necessarily the applicability and merit in the method suggested by the authors.
Author Response
Dr. Emmanuel Andrès Editor-in-Chief
Journal of Clinical Medicine
6 March 2023
Dear Dr Andrès
On behalf of my co-authors, I thank you for the opportunity to revise our manuscript (ID : jcm-2225008), titled “Feasibility of seated stepping and handshaking as a cardiopulmonary exercise testing: a pilot study”. The reviewers’ comments were very insightful and helped us in revising and improving our manuscript.
We have carefully addressed all the reviewers’ comments in our revised manuscript. We hope that our responses and revisions have adequately addressed the reviewers’ concerns and that the revised manuscript will now meet the high standards required for publication in your esteemed journal.
The manuscript has been rechecked and the necessary changes have been made in accordance with the reviewers’ suggestions. The responses to all comments have been prepared and given below.
Thank you for your consideration. We look forward to hearing from you.
Sincerely,
Yasuko Nishioka, MD, PhD
Department of Rehabilitation Medicine, Wakayama Medical University, 811-1 Kimiidera, Wakayama, 640-8509, Japan
Telephone: +81-73-447-0664
Fax: +81-73-446-6475
E-mail: kopandap627@yahoo.co.jp
Response to Reviewer 3 Comments
Manuscript ID: jcm-2225008
Type of manuscript: Brief Report
Title: Feasibility of seated stepping and handshaking as a cardiopulmonary exercise testing: a pilot study
Point-by-point responses to the reviewers’ comments
Dear Editor,
We would like to thank the Reviewers and Editors for their time and effort in reviewing our manuscript and for providing comments and suggestions, which have considerably helped us improve our manuscript. We have answered each of your points below and hope that our responses and revisions address all your comments.
Point 1: I commend the authors for the hard work and reverting with the revised manuscript in short time.
This is a very well written paper with sound analytical methods.
I can still see the novelty in this concept but not necessarily the applicability and merit in the method suggested by the authors.
Response 1: Our future challenge is to make SHS-CPX a clinically applicable test by increasing the number of subjects and including patients with various diseases so that SHS-CPX can be generalized.
